



# Assessment of NO₂ observations during DISCOVER-AQ and KORUS-AQ field campaigns

Sungyeon Choi[1,2], Lok N. Lamsal[1,3], Melanie Follette-Cook[1,4], Joanna Joiner[1], Nickolay A. Krotkov[1], William H. Swartz[5], Kenneth E. Pickering[1,6], Christopher P. Loughner[6,7], Wyat Appel[8], Gabriele Pfister[9], Pablo E. Saide[10], Ronald C. Cohen[11], Andrew J. Weinheimer[9], and Jay R. Herman[1,12]

[1]NASA Goddard Space Flight Center, Greenbelt, MD 20771, USA
[2]Science Systems and Applications, Inc., Lanham, MD 20706, USA
[3]Universities Space Research Association, Columbia, MD 21046, USA
[4]Morgan State University, Baltimore, MD 20251, USA
[5]Johns Hopkins University, Applied Physics Laboratory, Laurel, MD 20723, USA
[6]University of Maryland, College Park, MD 20742, USA
[7]NOAA Air Resources Laboratory, College Park, MD 20740, USA
[8]Environmental Protection Agency, Research Triangle Park, NC 27709, USA
[9]National Center for Atmospheric Research, Boulder, CO 80301, USA
[10]University of California, Los Angeles, CA 90095, USA
[11]University of California, Berkeley, CA 94720, USA
[12]University of Maryland Baltimore County, Baltimore, MD 21250, USA

**Correspondence:** Sungyeon Choi (sungyeon.choi@nasa.gov)

**Abstract.** NASA's Deriving Information on Surface Conditions from Column and Vertically Resolved Observations Relevant to Air Quality (DISCOVER-AQ) campaign in the United States and the joint NASA and National Institute of Environmental Research (NIER) Korea-United States Air Quality Study (KORUS-AQ) in South Korea were two field study programs that provided comprehensive, integrated datasets of airborne and surface observations of atmospheric constituents, including nitrogen

dioxide (NO₂), with a goal of improving the interpretation of spaceborne remote sensing data. Various types of NO₂ measurements were made, including in situ concentrations and column amounts of NO₂ using ground- and aircraft-based instruments, while NO₂ column amounts were being derived from the Ozone Monitoring Instrument (OMI) on the Aura satellite. This study takes advantage of these unique data sets by first evaluating in situ data taken from two different instruments on the same aircraft platform, comparing coincidently sampled profile-integrated columns from aircraft spirals with remotely sensed column

observations from ground-based Pandora spectrometers, intercomparing column observations from the ground (Pandora), aircraft (in situ vertical spirals), and space (OMI), and evaluating NO₂ simulations from coarse Global Modeling Initiative (GMI) and high-resolution regional models. We then use these data to interpret observed discrepancies due to differences in sampling and deficiencies in the data reduction process. Finally, we assess satellite retrieval sensitivity to observed and modeled a priori NO₂ profiles. Contemporaneous measurements from two aircraft instruments that likely sample similar air masses generally

agree very well but are also found to differ in integrated columns by up to 31.9%. These show even larger differences with Pandora, reaching up to 53.9%, potentially due to a combination of strong gradients in NO₂ fields that could be missed by aircraft spirals and errors in the Pandora retrievals. OMI NO₂ values are about a factor of two lower in these highly polluted





environments, due in part to inaccurate retrieval assumptions (e.g., a priori profiles), but mostly to OMI's areal ($> 312$ km$^2$) averaging.

**1 Introduction**

Nitrogen dioxide (NO$_2$) plays an important role in the troposphere by altering ozone production and OH radical concentration (Murray et al., 2012, 2014). It is one of the six United States Environmental Protection Agency (EPA) criteria pollutants because of its adverse health effects on humans (WHO, 2013). Major sources of nitrogen oxides (NO$_x$ = NO + NO$_2$) in the troposphere include combustion, soil, and lightning. As a trace gas with a relatively short lifetime, NO$_2$ is usually confined to

a local scale with respect to its source and therefore exhibits strong spatial and temporal variations, leading to difficulties in comparing NO$_2$ observations by methods with different atmospheric sampling.

Due to its distinct absorption features at ultraviolet/visible (UV/VIS) wavelengths, atmospheric NO$_2$ is observable from ground- and space-based remote sensing instruments. In particular, space-based measurements of tropospheric column NO$_2$ have been widely used to study spatial and temporal patterns (e.g., Beirle et al., 2003; Richter et al., 2005; Boersma et al.,

2008; Lu and Streets, 2012; Wang et al., 2012; Hilboll et al., 2013; Russell et al., 2010, 2012; Duncan et al., 2013; Lin et al., 2015), and long-term trends (e.g., van der A et al., 2008; Lamsal et al., 2015; Krotkov et al., 2016), and to infer NO$_x$ sources (e.g., Jaeglé et al., 2005; van der A et al., 2008; Bucsela et al., 2010; de Wildt et al., 2012; Lin, 2012; Ghude et al., 2010, 2013a; Mebust and Cohen, 2013; Pickering et al., 2016) and top-down NO$_x$ emissions (e.g., Martin et al., 2003; Konovalov et al., 2006; Zhao and Wang, 2009; Lin et al., 2010; Lamsal et al., 2011; Ghude et al., 2013b; Vinken et al., 2014; Schreier

et al., 2015; Cooper et al., 2017; Miyazaki et al., 2017; Liu et al., 2018). These observations have also been often used to assess chemical mechanisms (e.g., Martin et al., 2002; van Noije et al., 2006; Lamsal et al., 2008; Kim et al., 2009; Herron-Thorpe et al., 2010; Huijnen et al., 2010) as well as to infer the lifetime of NO$_x$ (e.g., Schaub et al., 2007; Lamsal et al., 2010; Beirle et al., 2011) in chemical transport models (CTMs). Surface NO$_2$ concentrations (Lamsal et al., 2008, 2014; Novotny et al., 2011; Bechle et al., 2013) and NO$_x$ deposition flux (Nowlan et al., 2014; Geddes and Martin, 2017) can also be estimated

using satellite NO$_2$ observations. As the accuracy of any applications of satellite data largely depends on the data quality, validation of satellite NO$_2$ observations is necessary.

A number of validation studies of space-based tropospheric NO$_2$ columns have been conducted using independent NO$_2$ observations from airborne in situ mixing ratio measurements (e.g., Boersma et al., 2008; Bucsela et al., 2008; Hains et al., 2010; Lamsal et al., 2014), ground-based total (e.g., Pandora instrument (Herman et al., 2009)) and tropospheric (e.g., MAX-

DOAS instrument (e.g., Vlemmix et al., 2010; Irie et al., 2012)) column measurements, and airborne high-resolution DOAS measurements (Lamsal et al., 2017; Nowlan et al., 2018). Most validation studies utilizing in situ/ground-based observations have reported that satellite measurements tend to underestimate tropospheric NO$_2$ columns, especially over highly polluted areas (e.g., Hains et al., 2010). Intrinsic limits of space-based measurements, however, pose a challenge in comparisons between satellite and in situ/ground-based measurements due to differences in representativeness. As stated above, NO$_2$ usually exhibits

very sharp spatial gradients (tens of meters to kilometers). In contrast, the spatial resolution of satellite measurements is too





coarse (tens of kilometers) to capture the fine spatial features of tropospheric $NO_2$ abundance. Therefore, it is important to recognize and account for the spatial variability while comparing satellite data with ground-based and in situ observations.

While the intrinsic resolution of satellite observations cannot be altered, there are ways to improve the derived satellite data products. The fidelity of the retrieved $NO_2$ product is dependent on the assumptions (e.g., $NO_2$ vertical profile shape, surface reflectivity) made in the retrieval algorithm. Some of the input parameters are available at much coarser resolution than the spatial resolution of OMI, introducing spatially (e.g., rural-to-urban) varying retrieval biases. Several studies show that the use of high-resolution $NO_2$ profiles results in significant improvements in retrievals (e.g., Russell et al., 2012; Lin et al., 2014; Lamsal et al., 2014; McLinden et al., 2014; Laughner et al., 2016, 2019; Goldberg et al., 2017). Deficiencies in model distributions of $NO_2$ may be identified and improved through rigorous evaluation with independent data, such as the suite of data collected during the Deriving Information on Surface Conditions from Column and Vertically Resolved Observations Relevant to Air Quality (DISCOVER-AQ) campaign deployments.

In this paper, we use comprehensive, integrated datasets of $NO_2$ gathered from surface, aircraft, and space instruments during NASA's DISCOVER-AQ and NASA's and NIER's Korea-United States Air Quality Study (KORUS-AQ) together with $NO_2$ model simulations to address questions regarding retrieval accuracy. We describe the datasets in Section 2.1 and the models in Section 2.2. As an example, we focus on the NASA Standard $NO_2$ Product from OMI onboard the Aura satellite and conduct retrieval studies using the algorithm as discussed in Section 2.3, but the approaches discussed here could be applied to similar products as well. Results are presented in Section 3.

## 2 Observations and Chemical Transport Models

### 2.1 $NO_2$ observations during DISCOVER-AQ and KORUS-AQ field campaigns

DISCOVER-AQ (https://www-air.larc.nasa.gov/missions/discover-aq/) and KORUS-AQ (https://www-air.larc.nasa.gov/missions/korus-aq/) were field study programs that provided comprehensive, integrated datasets of airborne and surface observations relevant to the diagnosis of surface air quality conditions from space. DISCOVER-AQ was a part of the NASA Earth Venture program and conducted four field deployments in Maryland (MD), California (CA), Texas (TX), and Colorado (CO) that covered different seasons and pollution regimes. KORUS-AQ was an international cooperation field study program conducted in the Republic of Korea (South Korea), sponsored by NASA and South Korean Government NIER. Table 1 summarizes the campaign locations and periods for the two field campaigns.

The primary objectives of DISCOVER-AQ and KORUS-AQ included (1) exploring the relationship between air quality at the surface and the tropospheric columns that can be derived from satellite orbit, (2) examining the diurnal variation of these relationships, and (3) characterizing the scales of variability relevant to the model simulation and remote observation of air quality. To accomplish these objectives, an observing strategy was designed to carry out systematic and concurrent in situ and remote sensing observations from a network of ground sites and research aircraft. The payloads on research aircraft consisted of several in situ instruments that differed minimally between campaigns. Ground-based trace gas observations included in situ surface and remote sensing Pandora measurements (Herman et al., 2009).





Figure 1 illustrates a conceptual view of the instruments and their sampling methods and areal coverage for NO$_2$ observa-
tions. While the aircraft (P-3B for DISCOVER-AQ and DC-8 for KORUS-AQ) make spirals (P-3B) or ascents/descents (DC-8)
over the site, the on-board NCAR and TD-LIF instruments measure in situ NO$_2$ profiles. The aircraft usually visit each site
2-4 times a day to observe the diurnal variations of the NO$_2$ profiles. Pandora and NO$_2$ ground monitor instruments are typi-
cally located at ground stations close to the aircraft profiles. Throughout the day, Pandora reports the total column NO$_2$ from
direct-sun measurements and the ground monitor reports the in situ surface NO$_2$ mixing ratio. Finally, OMI retrievals report a
tropospheric column NO$_2$ once a day in the afternoon; the OMI pixel has a much larger ground footprint as compared with the
in situ and Pandora measurements. Table 2 lists the sites with ground-based NO$_2$ monitors used in this analysis, along with the
type of instrument employed at each site and the numbers of aircraft profiles and Pandora measurements available from each
site near the time of OMI overpass. Detailed data descriptions follow in this section.

### 2.1.1 Vertical distribution of NO$_2$ by aircraft

In situ NO$_2$ volume mixing ratios (VMRs) were measured from the NASA P-3B (DISCOVER-AQ) and DC-8 (KORUS-AQ)
aircraft. The number of flights varied between campaigns, ranging from 10 for Texas to 22 for Korea. Flights took place during
a range of conditions, e.g., pollution episodes, clean days, weekdays, and weekends. Measurements usually commenced in the
morning and continued throughout the day with multiple sorties on a given day. During each sortie, the aircraft made vertical
spirals over surface sites, sampling NO$_2$ between ~300 m and 5 km from the Earth's surface. In Maryland, spirals were also
made over the Chesapeake Bay area, which did not have any ground monitors.

Airborne measurements were carried out using two different instruments and measurement techniques. The four-channel
chemiluminescence instrument from the National Center for Atmospheric Research (NCAR) measured NO$_2$ by photolysis
of NO$_2$ and subsequent chemiluminescence detection of NO$_2$ following oxidation of the photolysis product NO with ozone
(Ridley and Grahek, 1990). This instrument has an NO$_2$ measurement uncertainty of 10% and a 1-s, 2-sigma detection limit
of 50 parts per trillion by volume (pptv). We hereafter refer to these NO$_2$ measurements as "NCAR". The thermal dissocia-
tion laser-induced florescence (TD-LIF) method used by the University of Berkeley detects NO$_2$ directly and other nitrogen
species (e.g., total peroxynitrates, alkyl nitrates, HNO$_3$) following thermal dissociation of all oxides of nitrogen (NO$_y$) to NO$_2$
(Thornton et al., 2000). The laser-induced fluorescence method is highly sensitive for measuring NO$_2$, with a detection limit of
30 pptv. The measurement uncertainty is 5%. This instrument has a lower NO$_2$ sampling frequency than the NCAR instrument
due to its alternating measurement cycle for different species. We refer these NO$_2$ measurements as "TD-LIF".

Here we use 1-second merged data provided in the campaign data archives, and focus on early afternoon measurements
made within 1.5 h of the OMI overpass time (1:30 pm, approximately). Figure 2 shows the mean NO$_2$ profile for each of
the DISCOVER-AQ and KORUS-AQ campaigns. Measurements show considerable spatio-temporal variation as well as some
indication of a well-developed mixing layer with the maximum mixing ratio near to the ground. The mixing layer heights vary
with region and season. For example, in the MD campaign conducted in summer, the mixing layer stretches up to 800 hPa ( 2
km). In contrast, the mean profiles from the CA campaign conducted in winter show a shallow mixing layer extending only up
to 950 hPa (~700 m). Near-surface NO$_2$ mixing ratios also vary with the campaign locations and possibly with seasons with



highest near surface $NO_2$ in CA. In South Korea, the mean near-surface $NO_2$ mixing ratio is not as high as in CA, but a very high (~5 ppbv) $NO_2$ mixing ratio stretches up to 850 hPa, resulting in the greatest $NO_2$ column. While NCAR and TD-LIF

mean profiles generally agree with each other in the MD, CA, and CO campaigns, they exhibit larger differences in TX and South Korea.

### 2.1.2 In situ surface $NO_2$ measurements

To extend the altitude range of the vertical profiles discussed in Section 2.1.1, we merge in situ aircraft profile measurements with coincident in situ surface $NO_2$ measurements sampled over the duration of spirals (~20 minutes) by linearly interpolating

the $NO_2$ mixing ratios between the surface and the lowest aircraft altitudes. These new merged profiles contain a greater portion of the tropospheric $NO_2$ column. During both the DISCOVER-AQ and KORUS-AQ campaigns, in situ surface $NO_2$ monitors were deployed at several ground sites (Table 2). Measurements were carried out using one of four different types of $NO_2$ monitors, including chemiluminescence $NO_x$ monitor equipped with either molybdenum or photolytic converter, Cavity Attenuated Phase Shift (CAPS), and Cavity Ring-Down Spectroscopy (CRDS). The molybdenum converter analyzer measures

$NO_2$ indirectly by thermal conversion of $NO_2$ to NO using molybdenum and detection of NO by chemiluminescence that results from the reaction of NO with ozone. Since the reduction process could convert not only $NO_2$ but also other reactive nitrogen species, this instrument could overestimate $NO_2$ concentrations (Dunlea et al., 2007; Steinbacher et al., 2007; Lamsal et al., 2008; Dickerson et al., 2019). The magnitude of interference depends on the relative concentration of $NO_2$, nitric acid, alkyl nitrates, and peroxy-acetyl nitrate, which vary spatially, diurnally, and seasonally, and is difficult to quantify. Considering

their use in the sections below (Sections 2.3.2 and 3), we conducted a sensitivity study examining how 0–50% biases in molybdenum converter measurements could impact tropospheric columns derived from merged (aircraft + surface) profiles. We found that the errors are usually rather small at < 6% for various sites. Therefore, no attempt is made here to correct for the interference in these measurements, although we identify those sites in Table 2 and figures below.

The operating principle of a photolytic converter analyzer is also gas-phase chemiluminescence, but the use of a photolytic

converter to reduce $NO_2$ to NO makes it more specific to $NO_2$. As a result, this instrument provides nearly interference-free $NO_2$ measurements, with the exception of HONO (Ryerson et al., 2000). Measurement uncertainties for 1-hour averages are expected to be ~10% (Fehsenfeld et al., 1990).

The CAPS instrument detects $NO_2$ by measuring absorption around 450 nm. Baseline measurements spanning minutes to hours with a source of $NO_2$-free air are needed to determine $NO_2$ amounts. In contrast to the chemiluminescence/molybdenum

converter techniques, CAPS directly detects $NO_2$. Its specificity for $NO_2$ is affected by potential interference from species like glyoxal, water vapor, and ozone that absorb light within the bandpass of the instrument. The detection limit is <0.1 ppb for a 10-second measurement. $NO_2$ measurements from CAPS and chemiluminescence $NO_x$ monitors with molybdenum converter are reported to agree to within 2% (Kebabian et al., 2008).

CRDS is a sensitive and compact detector that measures multiple nitrogen species including $NO_2$. It employs a laser diode at

150 405 nm for direct detection of $NO_2$. Interferences arising from absorption by other trace gases, such as ozone and water vapor, are expected to be small. The measurement precision is 20 ppt at a 1-second time resolution and the accuracy is better than 5%,



which is primarily limited by the NO$_2$ absorption cross-section used in the data reduction process. The total reactive nitrogen (NO$_y$) measured by CRDS and chemiluminescence NO$_x$ monitor with molybdenum converter is found to agree to within 12% (Wild et al., 2014).

### 2.1.3 Pandora total column NO$_2$

In addition to in situ measurements, each campaign hosted ground-based networks of Pandora instruments. Pandora is a small, commercially available sun-viewing spectrometer optimized for detection of trace gases, including NO$_2$. It measures direct solar spectra in the 280-525 nm spectral range with 0.6 nm resolution. A detailed description of the instrument's design, operation, and retrieval method can be found in Herman et al. (2009, 2018). The NO$_2$ retrieval algorithm includes (1) a direct-sun spectral fitting method similar to traditional Differential Optical Absorption Spectroscopy (DOAS) (Platt, 1994) using one measurement (or an average of several measurements) as a reference spectrum to derive relative NO$_2$ slant column densities (SCDs), (2) application of the Modified Langley Extrapolation (MLE) to derive total NO$_2$ SCDs, and (3) conversion of total NO$_2$ SCDs to vertical column densities (VCDs) using the direct sun air mass factor (AMF) as follows:

$$VCD = SCD/AMF \qquad (1)$$

The spectral fitting is performed over the 400-440 nm window; it fits NO$_2$ cross sections at 254.5 K (Vandaele et al., 1998), ozone (Brion et al., 1993) and a 4th order smoothing polynomial, and applies a wavelength shift and a constant offset. In clear-sky conditions, this instrument provides total NO$_2$ VCD with precision of $2.7 \times 10^{14}$ molec cm$^{-2}$ and an absolute accuracy of $1.3 \times 10^{15}$ molec cm$^{-2}$ (Herman et al., 2018). Potential sources of error in NO$_2$ retrievals include calibration of raw data, chosen reference spectrum, and the use of a fixed temperature for the NO$_2$ cross-section. Pandora NO$_2$ data have been compared with data from direct-sun Multi-Function DOAS (MFDOAS) and Fourier Transform Ultraviolet Spectrometer (UVFTS) (Herman et al., 2009) and have been found to agree within 12%. These data are regularly used to validate satellite NO$_2$ retrievals (e.g., Lamsal et al., 2014; Tzortziou et al., 2015, 2018; Ialongo et al., 2016).

Here, we use clear-sky quality controlled (root-mean-square (rms) < 0.05 and errors < 0.05 DU) 80-sec total NO$_2$ column data averaged over the duration of each aircraft spiral. We infer tropospheric column NO$_2$ by subtracting the OMI stratospheric column from the Pandora total column to compare with tropospheric NO$_2$ from in situ and OMI observations.

## 2.2 NO$_2$ simulations

### 2.2.1 GMI simulation

The Global Modeling Initiative (GMI) 3-Dimensional chemical transport model (CTM) simulates the troposphere and stratosphere (Strahan et al., 2013) with a stratosphere-troposphere chemical mechanism (Duncan et al., 2007) updated with the latest chemical rate coefficients (Burkholder et al., 2015) and time-dependent natural and anthropogenic emissions (Strode et al., 2015). Aerosol fields are computed on-line with the Goddard Chemistry Aerosol Radiation and Transport (GOCART) model (Chin et al., 2014, and references therein). Tropospheric processes such as NO$_x$ production by lightning, scavenging,



and wet and dry deposition are also represented in the model. The GMI simulations used in this work were constrained with meteorology from the Modern-Era Retrospective Analysis for Research and Applications, Version 2 (MERRA-2) meteoro-

logical fields (Gelaro et al., 2017) at 72 vertical levels from the surface to 0.01 hPa, with a resolution ranging from ~150 m in the boundary layer to ~1 km in the upper troposphere and lower stratosphere, and at a horizontal spatial resolution of $1.25°$ longitude$\times 1.0°$ latitude.

GMI simulations have been evaluated in the troposphere and stratosphere. Strode et al. (2015) showed good agreement with tropospheric $O_3$ and $NO_x$ trends in the U.S. in a 1990–2013 hindcast simulation. Strahan et al. (2016) demonstrated realistic

seasonal and interannual variability of Arctic composition using comparisons to Aura MLS $O_3$ and $N_2O$. The simulation of $NO_2$ in both the troposphere (Lamsal et al., 2014) and stratosphere (Spinei et al., 2014; Marchenko et al., 2015) have been shown to be in good agreement with independent measurements. We sample the model profile at the times and locations of airborne measurements. Figure 2 compares GMI $NO_2$ profiles with collocated aircraft measurements during the DISCOVER-AQ and KORUS-AQ field campaigns. The GMI simulation generally captures the vertical distribution of $NO_2$ in the free-

troposphere, is somewhat lower in the middle and upper parts of the mixing layer, and exhibits sharper gradients between the boundary layer and surface. Due to the coarse spatial resolution of the GMI model, the surface pressure of the GMI profiles differs from the measurements, especially over complex terrain in CA, CO, and Korea.

### 2.2.2 $NO_2$ simulations using regional models

For each DISCOVER-AQ and KORUS-AQ deployment, a high-resolution model simulation was conducted. We use $NO_2$

profiles from those simulations to examine their effect on retrievals in Section 2.3.2 and to downscale OMI $NO_2$ retrievals in Section 2.3.3. Below we provide a brief description of each simulation. Information about model options for these simulations can be found in Table A1 in the Appendix. For most of the campaigns, the near-surface $NO_2$ concentration and the model profile shapes agree in general with the NCAR and TD-LIF profiles. In TX, however, the CMAQ simulation shows lower mixing ratios than observations throughout the mixing layer (Figure 2).

**MD:** The Weather Research and Forecasting (WRF) model was run (Loughner et al., 2014) from May 24, 2011 through August 1, 2011 at horizontal resolutions of 36, 12, 4, and 1.33 km with 45 vertical levels from the surface to 100 hPa with 16 levels within the lowest 2 km. Meteorological initial and boundary conditions were taken from the 12 km North American Mesoscale (NAM) model. Output from the 4 and 1.33 km WRF simulations were fed into the Community Multiscale Air Quality (CMAQ; Byun and Schere (2005)). Chemical initial and boundary conditions for the 4 km CMAQ run came from a

12 km CMAQ simulation covering the continental US, which was performed for the GEO-CAPE Regional Observing System Simulation Experiment (OSSE). The creation of the emissions used within the CMAQ simulation is described in Loughner et al. (2014) and Anderson et al. (2014). CMAQ was run with reduced mobile emissions by 50% and an increase in the photolysis frequency of NTR based on Anderson et al. (2014).

**CA:** The coupled WRF-CMAQ modeling system (Wong et al., 2012) was run from January 1, 2013 through February

28, 2013 (2013 DISCOVER-AQ California campaign period) at horizontal resolutions of 4 and 2 km, with 35 vertical levels from the surface to 50 hPa and an average height of the middle of the lowest layer of 20 m. WRF version 3.8 and CMAQ





version 5.2.1 were used in a coupled format, allowing for frequent communication between the meteorological and chemical transport models and indirect effects from aerosol loading on the meteorological calculations in WRF. Meteorological initial and boundary conditions were taken from the 12 km NAM reanalysis product from NOAA. Observation nudging above the
planetary boundary layer (PBL) using four-dimensional data assimilation (FDDA) was applied in WRF. Chemical initial and boundary conditions for the 4 km CMAQ simulation came from a 12 km CMAQ simulation covering the continental US, while initial and boundary conditions for the 2 km simulation were obtained from the 4km WRF-CMAQ simulation. Emissions are based on the 2011 U.S. National Emissions Inventory (NEI) with year-specific updates to point and mobile sources, while biogenic emissions were calculated in-line in CMAQ using the Biogenic Emissions Inventory System (BEIS).

**TX:** To simulate the DISCOVER-AQ Texas campaign, a WRF model simulation was performed from August 18, 2013 through October 1, 2013, covering the entire field deployment in September 2013. The model was run at 36, 12, and 4 km, and 1.33 horizontal resolutions with 45 levels from the surface to 50 hPa. Meteorological initial and boundary conditions were taken from the 12 km North American Mesoscale (NAM) model. Output from the 4 km and 1.33 km simulations were used to run the CMAQ model. Chemical and initial boundary conditions for the outer domain were taken from the Model for Ozone and
Related chemical Tracers (MOZART) Chemical Transport Model (CTM). Detailed information about these simulations and the emissions used can be found at http://aqrp.ceer.utexas.edu/projectinfoFY14_15/14-004/14-004%20Final%20Report.pdf.

**CO:** For the Colorado deployment, WRF was run from July 9, 2014 through August 20, 2014 at spatial resolutions of 12 km (covering the Western US) and 4 km (covering Colorado). The model top was set at 50 hPa, with 37 levels in the vertical. Analysis fields from the European Centre for Medium-Range Weather Forecasts (ECMWF) were used for meteorological
initial and boundary conditions. Chemical initial and boundary conditions for the outer domain were taken from Real Time Air Quality Monitoring System (RAQMS) model output. Further information about this simulation can be found at https://www.colorado.gov/airquality/tech_doc_repository.aspx?action=open&file=FRAPPE-NCAR_Final_Report_July2017.pdf.

**Korea:** Air quality forecasts were performed using the Weather Research and Forecasting model (Skamarock et al., 2008) coupled to Chemistry (WRF-Chem) (Grell et al., 2005) model to support KORUS-AQ flight planning and post-campaign
analysis. The modeling domains consist of a regional domain of 20 km resolution covering major sources of transboundary pollutants affecting the Korean Peninsula: anthropogenic pollution from eastern China, dust from inner China and Mongolia, and wild fires from Siberia (Saide et al., 2014). A 4 km resolution domain was nested and covered the Korean Peninsula and surroundings, which encompassed the region where the DC-8 flights were planned and better resolved local sources. Anthropogenic emissions were developed by Konkuk University for KORUS-AQ forecasting and are described in Goldberg
et al. (2019).

## 2.3 OMI NO$_2$ observations

The Ozone Monitoring Instrument (OMI) aboard the NASA Aura satellite provides measurements of solar backscatter that are used to retrieve total, stratospheric, and tropospheric NO$_2$ columns with a native ground resolution varying from 13 km×24 km near nadir to 40 km×250 km at swath edges (Levelt et al., 2006, 2018). The Aura satellite was launched on 15 July 2004
into a Sun synchronous polar orbit with a local equator crossing time of  13:45 in the ascending node. OMI is one of the most





stable UV/Vis satellite instruments providing a long-term high-resolution data record with low degradation (Dobber et al., 2008; DeLand and Marchenko, 2013; Schenkeveld et al., 2017).Since the middle of 2007, an anomaly began to appear in OMI radiances in certain rows affecting all Level 2 products (Schenkeveld et al., 2017). This "row anomaly" can be easily identified and the affected rows are discarded.

### 255    2.3.1    Standard OMI NO$_2$ product

Here we use the Standard OMI NO$_2$ product (OMNO2), version 3.1, with updates from version 3.0 (Krotkov et al., 2017). The NO$_2$ retrieval algorithm uses the differential optical absorption spectroscopy (DOAS) technique. The retrieval method includes (1) determination of NO$_2$ slant column density (SCD) using a DOAS spectral fit of the NO$_2$ cross-section from measured reflectance spectra over the 402-465 nm range; (2) calculation of an air mass factor (AMF) that is required to convert SCD into vertical column density (VCD); and (3) a scheme to separate stratospheric and tropospheric VCDs. The AMF calculation is performed by combining NO$_2$ measurement sensitivity (scattering weights) from the TOMS RADiative transfer model (TOMRAD, Dave (1964)) with the a priori relative vertical distribution (profile shape) of NO$_2$ taken from the GMI CTM. Computation of scattering weights requires information on viewing and solar geometries, terrain and cloud reflectivities, terrain and cloud pressures, and cloud cover (radiative cloud fraction).

The version used here represents a significant advance over previous versions (Bucsela et al., 2006, 2013; Celarier et al., 2008; Lamsal et al., 2014). It includes an improved DOAS algorithm for retrieving slant column densities (SCDs) as discussed in Marchenko et al. (2015). The key features of the algorithm include more accurate wavelength registration between Earth radiance and solar irradiance spectra, iterative accounting of rotational Raman scattering effect, and sequential SCD retrieval of NO$_2$ and interfering species (water vapor and glyoxal). Solar irradiance reference spectra are monthly average data derived from OMI measurements instead of an OMI composite solar spectrum used in prior versions. Cloud pressure and cloud fraction are taken from an updated version of the OMCLDO2 cloud product that includes updated look-up tables and O$_2$-O$_2$ SCD retrieved with a temperature correction (Veefkind et al., 2016). A priori NO$_2$ profiles are as discussed in Lamsal et al. (2015) and Krotkov et al. (2017) and use 1° latitude×1.25° longitude GMI model-based monthly a priori NO$_2$ profiles with year-specific emissions. This retrieval version also uses more accurate information on terrain pressure that is calculated from high resolution Digital Elevation Model (DEM) data at 3 km resolution and GMI terrain pressure.

### 2.3.2    Re-calculation of OMI NO$_2$ AMF using alternative NO$_2$ profiles

NO$_2$ vertical profiles, especially in the troposphere, vary strongly in both space and time. The simulated NO$_2$ profiles from a global CTM (GMI) employed in the operational NO$_2$ retrieval, while offering a good option at a global scale, may not sufficiently capture the distribution of NO$_2$ at OMI's ground resolution. Using pre-calculated scattering weights (Sw) made available in the OMNO2 product and alternative information on vertical NO$_2$ profile shape (Xa), the OMI NO$_2$ AMF can be readily re-calculated (Lamsal et al., 2014).

$$AMF_{trop} = \frac{\sum_{surface}^{tropopause} Sw \cdot Xa}{\sum_{surface}^{tropopause} Xa}, \tag{2}$$





where the integral from surface to the tropopause yields the tropospheric AMF (AMF$_{\text{trop}}$). Scattering weights vary with viewing/solar geometry, cloud/aerosol conditions, and surface reflectivity, but they are assumed to be independent of the vertical

distribution of NO$_2$. The typical vertical distribution of scattering weights is characterized by lower values in the troposphere due to reduced sensitivity owing to Rayleigh scattering and higher values (corresponding to a nearly geometric AMF) in the stratosphere. The AMF is therefore highly sensitive to NO$_2$ profile shape in the lower troposphere.

Here, we investigate how a priori NO$_2$ profiles affect OMI tropospheric AMF and consequently the retrieval of OMI tropospheric NO$_2$ VCD. For this, we combine the measured profile (from surface to ~5 km) with coincidently sampled simulated

NO$_2$ from GMI (5 km to tropopause) to create a complete tropospheric NO$_2$ profile. We choose the GMI simulation over the high resolution model simulations because we found that the GMI generally better performed in the free-troposphere as compared to the regional models. We then interpolate the pressure-tagged NO$_2$ observations (aircraft NCAR NO$_2$ + surface) onto the pressure grid of the OMI NO$_2$ scattering weight. The tropospheric AMFs obtained using individual measured profiles (AMF$_{\text{obs}}$) are compared with the AMFs in the OMI standard product (AMF$_{\text{SP}}$), which are calculated using the GMI yearly

varying monthly climatology (Figure 3a). AMF$_{\text{SP}}$ is generally higher than AMF$_{\text{obs}}$ by 34% on average, with the largest difference (61.6%) for TX and the smallest difference (16.6%) for Korea; this means that the OMI SP VCDs, based on the AMF$_{\text{SP}}$, are corresponding smaller on average than the those based on measured profiles. The correlation ranges from fair (r = 0.41, N = 21) for MD and TX to excellent (r $\geq$ 0.92, N = 36) for CA and Korea.

To explore how NO$_2$ profiles from high-resolution model simulations could affect OMI NO$_2$ retrievals, we calculate tro-

pospheric AMFs using simulated monthly NO$_2$ profiles (AMF$_{\text{HR}}$). Since the OMI ground pixel size is much larger than the model grid boxes, we derive an average profile of all model grid boxes located within one OMI pixel and use it to calculate AMF$_{\text{HR}}$. Figure 3(b) compares AMF$_{\text{obs}}$ with AMF$_{\text{HR}}$; it suggests improved agreement as compared to AMF$_{\text{SP}}$ (Figure 3a) especially for CA, CO, and Korea, albeit with no significant improvement in the correlation.

We also considered how using AMFs based on monthly mean profiles, such as the OMI SP, impacts retrieved NO$_2$. To assess

this, we calculated AMFs using both daily (AMF$_{\text{obs}}$) and campaign-average measured NO$_2$ profiles (AMF$_{\text{obs}-\text{m}}$). Figure 3(c) shows that AMF$_{\text{obs}}$ and AMF$_{\text{obs}-\text{m}}$ agree to within 5.3% and exhibit excellent correlation (r > 0.8). That is, the use of a mean profile does not make a significant difference compared to the individual daily profiles, implying that the average profile generally captures the local vertical distribution fairly well. Somewhat larger scatter in TX may be related to stronger land-sea breeze dynamics that could affect the vertical distribution of NO$_2$ in both the boundary layer and free-troposphere. Our results

here differ with previous studies that reported improved agreement of OMI NO$_2$ retrievals using simulated daily NO$_2$ profiles with independent observations (Valin et al., 2013; Laughner et al., 2019), although Laughner et al. (2019) also suggested poorer performance with daily profiles in the southeast US than in other regions.

### 2.3.3 Downscaled OMI NO$_2$ data

The NO$_2$ value associated with an OMI ground pixel is averaged over a large area. This spatial smoothing leads to a loss of

information on sub-pixel variation, which could be considerable for NO$_2$ especially over urban source regions. Therefore, it



is important to recognize and address this limitation while assessing, interpreting, and using satellite $NO_2$ data. Here we use high-resolution $NO_2$ model simulations for sub-pixel variation.

We apply the method described by Kim et al. (2016, 2018) to downscale OMI $NO_2$ retrievals, which are then compared with aircraft and Pandora data. This method applies high resolution model-derived spatial-weighting kernels to individual

OMI pixels and calculates sub-pixel variability within the pixel. The major assumption is that the model captures the spatial distribution of emission sources and $NO_2$ transport patterns well. The method ensures that the quantity (total number of molecules) of the satellite data over the pixel is numerically preserved, while adding higher resolution spatial information to the derived tropospheric $NO_2$ columns.

Figure 4 illustrates the downscaling of tropospheric $NO_2$ for an OMI pixel using the high resolution CMAQ simulation over

Essex, Maryland. The tropospheric $NO_2$ column observed by OMI ($5.9 \times 10^{15}$ molec cm$^{-2}$) is 25.7% higher than the average of the CMAQ $NO_2$ columns over the pixel. The spatial weighting kernels suggest more than an order of magnitude difference in $NO_2$ within this single OMI pixel. Applying the kernels to the original OMI pixel value results in a range of sub-pixel $NO_2$ column values from $1.9 \times 10^{15}$ molec cm$^{-2}$ over a clean background to $3.2 \times 10^{16}$ molec cm$^{-2}$ over a polluted hotspot.

Figure 5 demonstrates how the downscaled OMI $NO_2$ data using high-resolution $NO_2$ output from a CMAQ simulation

compares with the original OMI $NO_2$ data from the standard product. Both OMI SP and CMAQ show enhanced $NO_2$ columns at major urban areas, but their magnitudes differ, with OMI showing lower values. As described above, OMI's field of view covers a large area, sampling the $NO_2$ field over the entire pixel, while the actual $NO_2$ distribution (better resolved by the CMAQ simulation) is defined by local source strengths, chemistry, and wind patterns that can occur at much finer spatial scales. By employing the relative ratios inside an OMI pixel rather than the overall magnitude of simulated columns, the

downscaling technique yields a more detailed structure, enhancing $NO_2$ over sources and dampening elsewhere by more than a factor of two.

## 3   Results and Discussion

### 3.1   Comparison between in situ observations

Figure 6a and Table 3 summarize how the two airborne in situ $NO_2$ tropospheric column measurements compare. We derive the

column amount by first extending the NCAR and TD-LIF $NO_2$ profiles to the same surface $NO_2$ concentration measurements and then integrating the $NO_2$ profiles. The only exception is at the Chesapeake Bay of the MD campaign, the only marine site used in this study; we extend a constant $NO_2$ mixing ratio measured at the lowest aircraft altitudes to the surface. To compare with OMI and Pandora retrievals, $NO_2$ amounts for the missing portion from the top of aircraft altitude to the tropopause are added from the GMI simulation. This amount varied between $4.7 \times 10^{14}$ molec cm$^{-2}$ and $1.2 \times 10^{15}$ molec cm$^{-2}$ and

represented an average 5% of the tropospheric $NO_2$ columns but can reach up to 50.8% for an individual profile. Overall, the two airborne in situ columns generally agree very well and exhibit excellent correlation (r = 0.87–0.99). The correlation and mean difference differ among the five campaigns, with TD-LIF higher than NCAR by 31.9% in TX and 11.6% in Korea but lower by ~10% in MD and CO. The observed difference in TX is much larger than the reported uncertainty of both NCAR and





TD-LIF measurements. Analysis of individual profiles suggests that the data from TD-LIF are generally higher than NCAR at
all altitudes, regardless of the $NO_2$ pollution level (Figure 7). The underlying cause of this difference is not clear, but it may be
associated with the applied calibration standard or an interference issue for either or both of the two measurements. The small
difference elsewhere could come from the lower measurement frequency of TD-LIF as compared with the NCAR instrument.

### 3.2    Comparison between Pandora and aircraft observations

Figures 6b-6c and Table 3 show the comparison between the Pandora and the two airborne tropospheric $NO_2$ column mea-
surements. We derive tropospheric columns from Pandora by subtracting collocated OMI stratospheric $NO_2$ columns from the
Pandora total $NO_2$ column retrievals. The relationship between the aircraft and Pandora data is not as good as between the two
aircraft measurements themselves. The correlation ranges from fair (r = 0.42) to excellent (r = 0.95) for NCAR versus Pandora
and poor (r = 0.18) to excellent (r = 0.94) for TD-LIF versus Pandora, with higher correlation in CO, TX, and Korea and
lower correlation in MD and CA. Pandora data are about a factor of two lower than aircraft measurements in TX. Elsewhere,
Pandora data agree with aircraft measurements to within 20% on average, although much larger differences are observed for
individual sites. A larger discrepancy for Pandora data in TX is also reported by Nowlan et al. (2018), who used various $NO_2$
measurements to evaluate Geo-TASO $NO_2$ retrievals. Reasons for such exceptionally large differences could include strong
gradients in the $NO_2$ field that are missed by aircraft spirals, errors in Pandora retrievals, or both.

### 3.3    Assessment of OMI $NO_2$ retrievals

We compare OMI tropospheric $NO_2$ columns with Pandora data and vertically integrated columns from aircraft spiral at 23
locations (Table 2) during the DISCOVER-AQ and KORUS-AQ field campaigns. Collocated aircraft and Pandora data are
temporally matched to OMI by allowing only the measurements made within 1.5 h of the OMI overpass time. We infer
tropospheric columns from Pandora by subtracting OMI-derived stratospheric $NO_2$ from Pandora total columns. For OMI, we
include quality-controlled, cloud-free (cloud radiance fraction < 0.5) data from all cross-track positions, but exclude the data
affected by the row anomaly.

Figure 8 (a and b) and Table A2 present tropospheric $NO_2$ columns from the OMI standard product compared with integrated
columns from NCAR and TD-LIF instruments. Although the OMI and aircraft data are significantly correlated (r = 0.39–0.87),
OMI $NO_2$ retrievals are generally lower, with largest difference in CO and smallest difference in MD. OMI data are also
lower than Pandora as shown in Figure 8c. The magnitude of the difference and the degree of correlation with OMI vary
for NCAR, TD-LIF, and Pandora measurements. This discrepancy between OMI, aircraft spiral columns, and Pandora's local
measurements is due to a combination of strong $NO_2$ spatial variation, size of OMI pixels, and the placement of the sites, but
OMI retrieval errors arising from inaccurate information in the AMF calculation, such as a priori $NO_2$ profiles, and potential
errors in the validation sources themselves also contribute.

Figure 8(d-f) and Table A3 show the comparison after partially accounting for OMI retrieval errors arising from a priori
$NO_2$ profiles taken from the GMI model. Replacing the model profiles with the NCAR and TD-LIF observed $NO_2$ profiles in
the AMF calculations addresses the issues related to model inaccuracies, although the measured profiles may not necessarily



represent the true average NO$_2$ over the entire OMI pixel (e.g., Figure 4). Nevertheless, using observed profiles reduces OMI's mean differences with NCAR by 8%–29.2%, TD-LIF by 8.7%–24.4%, and Pandora by 6.8%–24.2%. Changes are largest in TX and smallest in CA and Korea. Correlations are either improved or remain similar.

Figure 8(g-i) and Table A4 show the comparison of OMI NO$_2$ columns derived using observed profiles with NCAR, TD-LIF, and Pandora observations after accounting for spatial variation in the NO$_2$ field as suggested by the CMAQ simulation. After downscaling, agreement of OMI NO$_2$ columns improves further with NCAR by 1.1%–41.5%, TD-LIF by 1.2%–39.7%, and Pandora by 1.2%–33.2%. Exceptions are MD for both aircraft and Pandora data, and TX for Pandora data only. Changes are small in MD and Korea and large in CA and TX. The larger difference in TX is due to significant underestimation of NO$_2$ by Pandora instruments. The correlation improves in MD and TX, but reduces in CA, CO, and Korea. These results suggest that downscaling helps explain some of the discrepancies between OMI, aircraft, and Pandora observations. Variations among campaign locations may also point to difficulty related to the fidelity of the CMAQ simulations.

Figure 9 summarizes the comparison of OMI with aircraft and Pandora measurements. Here we present site mean columns observed from all measurements during the entire campaign periods. OMI captures the overall spatial variation in site means. In relatively cleaner places (NO$_2$ VCD $\leq 5 \times 10^{15}$ molec cm$^{-2}$), OMI agrees well with NCAR and TD-LIF columns. OMI values are generally lower in polluted areas.

### 3.4 Implications for satellite NO$_2$ validations

NO$_2$ measurements from a variety of instruments and techniques conducted during the DISCOVER-AQ and KORUS-AQ field deployments provided a unique opportunity to assess correlative data and realize the strengths and limitations of the various measurements. Some of the techniques are still in a state of development and evaluation, and the data have not been fully validated. Additional complications arise when comparing measurements covering different areal extent. This is particularly true for a short-lived trace gas like NO$_2$ that has a large spatial gradient, especially in the boundary layer.

The NCAR and TD-LIF instruments on board the same aircraft (P-3B during DISCOVER-AQ and DC-8 during KORUS-AQ) offer valuable insights on vertical distribution of NO$_2$, a critical piece of information needed for satellite retrievals. Despite their adjacent locations on the aircraft, they did not sample the same air mass throughout each profile due to their different NO$_2$ measurement frequencies. Despite this, and even using independent measurement techniques with unique sources of uncertainties, NO$_2$ measurements from the two instruments exhibit excellent correlation and very good agreement in most cases. However, varying discrepancies between the two instruments among campaigns with campaign-average differences reaching up to 31.9% is unlikely to be related solely to the sampling issues, but rather to issues pertaining to measurement methods. It is crucial to reconcile these differences and improve the accuracy of these measurements for meaningful validation and improved error characterization of satellite NO$_2$ retrievals.

In situ aircraft spirals miss significant portions of the tropospheric NO$_2$ column, especially from the ground to the lowest level of aircraft altitude, typically 200-300 m above ground level. In this analysis, we account for the missing portion above the aircraft profile by using coincidently sampled simulated NO$_2$ profiles. For the portion below the aircraft profile we extrapolate to surface monitor data. The latter step can be a significant error source, given that it assumes spatial homogeneity





over the spiral domain. Additional errors could come from the use of different types of monitors that were deployed during the DISCOVER-AQ and KORUS-AQ campaigns (see Section 2.1.2). In particular, $NO_2$ data from molybdenum converter analyzers are biased high by variable amounts that are difficult to quantify and correct (e.g., Lamsal et al., 2008). Use of more accurate $NO_2$ monitors, such as photolytic converter analyzers, together with balloon-borne $NO_2$ sondes (Sluis et al., 2010) of

similar accuracy would complement in situ aircraft profiles.

While total column $NO_2$ retrievals from the ground-based remote sensing Pandora instrument are useful to track temporal changes, their use for satellite validation or for comparing with aircraft spiral data can be onerous particularly over locations with large $NO_2$ spatial gradients, such as cities. Pandora's field of view is so narrow that it serves as a point measurement. Additionally, Pandora data are subject to retrieval errors arising predominantly from the use of an incorrect reference spectrum

as well as fixed temperature for the $NO_2$ cross-section in the spectral fitting procedure. Failure to apply a reference spectrum derived using weeks of measurements from the same site often yields systematic biases in the retrieved $NO_2$ columns. Improved calibration and data processing are therefore needed to improve the Pandora data quality. Concurrent spatial $NO_2$ observations from other ground-based (e.g., Multi Axis Differential Optical Absorption Spectroscopy (MAX-DOAS), Vlemmix et al., 2010) or airborne (e.g., Geostationary Trace gas and Aerosol Sensor Optimization (GeoTASO), Nowlan et al., 2016) platforms would

facilitate inter-comparison among measurements of different spatial scales.

Validation of $NO_2$ observations any satellite instrument, including OMI, is complicated by a variety of factors, principally the ground area covered by the instrument's field of view. As discussed in Section 3.3, disagreement between partially (spatially and temporally) matched OMI $NO_2$ and validation measurements made near sources may be reasonably anticipated, and ought to be expected. Therefore, it may be necessary to use a proper validation strategy, such as downscaling of satellite data using

either observed or modeled $NO_2$ as presented in Figure 8(g-i) and Table A4. It also underscores the need of comprehensive high quality long-term observations for validation. Enhanced agreement with OMI retrievals revised using observed $NO_2$ profiles is indicative of retrieval errors from model-based a priori vertical $NO_2$ profile shapes (Figure 8(d-f), Table A3), and highlights the need of approaches to address the issue. Moreover, improved accuracy in other retrieval parameters, both surface and atmospheric, help enhance the quality of satellite $NO_2$ retrievals (Laughner et al., 2019; Vasilkov et al., 2017, 2018; Lorente

et al., 2018; Lin et al., 2014, 2015; Liu et al., 2019; Noguchi et al., 2014; Zhou et al., 2011)

## 4    Conclusions

We conducted a comprehensive inter-comparison among various $NO_2$ measurements made during the five field deployments of DISCOVER-AQ and KORUS-AQ. The field campaigns were conducted in four US states (Maryland, California, Texas, and Colorado), and South Korea. The analyzed data sets were obtained from surface monitors, the NCAR and TD-LIF airborne

instruments, ground-based Pandora instruments, and space-based OMI. We investigated the data from 23 sites among the 5 campaigns, when measurements from all these instruments were available. We focused on analysis of tropospheric $NO_2$ column amounts. $NO_2$ mixing ratio measurements from the surface monitors and airborne instruments were merged and integrated





to yield tropospheric columns while the Pandora tropospheric columns were obtained by subtracting the OMI stratospheric column from Pandora total column observations.

In order to compare OMI $NO_2$ tropospheric columns with the available validation measurements, we used a combination of observed and simulated $NO_2$ vertical profiles to re-calculate tropospheric $NO_2$ columns using the OMI Standard Product (OMNO2), version 3.1. To overcome the challenge of comparing OMI $NO_2$ with its relatively large pixel size to the airborne/ground-based measurements with small spatial scales, we additionally applied a downscaling technique, whereby OMI tropospheric $NO_2$ columns for each ground pixels are downscaled using high resolution CMAQ (DISCOVER-AQ) or

WRF-Chem (KORUS-AQ) model simulations. Therefore, the comparisons here include three kinds of OMI $NO_2$ tropospheric columns: (1) OMI Standard Product, (2) OMI data re-calculated using observed $NO_2$ profiles, and (3) downscaled OMI $NO_2$ data.

The tropospheric columns from the NCAR and TD-LIF airborne instruments generally show good agreement with a mean difference of 8.4% and correlation coefficients in the 0.87–0.99 range. The Pandora columns also agree variably with the two

airborne instruments, with the campaign average difference in the range of 3% to 54%, but the correlation is not as good (r = 0.18–0.95) as between the two airborne instruments themselves. There are differences among the campaigns. In particular, all three instruments show the largest discrepancies in the TX campaign; TD-LIF is higher than NCAR by ~31.9%, and Pandora data are lower by ~39% and ~54% as compared to NCAR and TD-LIF measurements, respectively.

All three OMI $NO_2$ columns (Standard Product, based on observed $NO_2$ profiles, and downscaled) exhibit good correla-

tion with the airborne/ground-based measurements. In terms of quantitative agreement, the OMI SP column is smaller than airborne/ground-based measurements. Retrievals using observed $NO_2$ profiles bring the OMI column closer to validation measurements. Applying downscaling to OMI data provides further improvement in agreement, albeit little or insignificant change in correlation perhaps due to the use of model simulations for downscaling.

As discussed in section 3.3, disagreement between the comparatively large OMI pixel and smaller scale ground and aircraft

measurements is to be expected due to the large spatial variability of $NO_2$. Techniques such as the downscaling method shown here can reduce this discrepancy. However, robust evaluation of $NO_2$ tropospheric column retrievals is further confounded by the current lack of agreement among ground based and in-situ measurements. Future validation strategies for satellite observations of tropospheric column $NO_2$ will need to address these differences.

*Data availability.* Airborne and ground-based, and Pandora $NO_2$ data during the DISCOVER-AQ and KORUS-AQ campaigns are available

at the NASA Langley's campaign data web archive (https://www-air.larc.nasa.gov/index.html). OMI $NO_2$ Standard Product (SP) data are available at NASA Goddard Earth Sciences Data and Information Services Center (GES DISC) (https://disc.gsfc.nasa.gov).

**Appendix A**

This section includes Table A1–A4.



*Author contributions.* SC, LL, JJ, NAK, MFC, WHS, and KEP designed the data analysis. CPL, WA, GP, and PES provided the model
simulations. RCC and AJW provided the airborne in situ measurements. JRH provided the ground-based Pandora measurements. SC, LL,
MFC, WHS, CPL, WA and PS wrote the manuscript with comments from all coauthors.

*Competing interests.* The authors declare that no competing interests are present.

*Acknowledgements.* The work was supported by NASA's EarthScience Division through an Aura Science team and Atmospheric Compo-
sition Modeling and Analysis Program (ACMAP) grants. The Dutch–Finnish-built OMI instrument is part of the NASA EOSAura satellite
payload. The OMI instrument is managed by KNMI and the Netherlands Agency for Aero-space Programs (NIVR). P. E. Saide would like to
acknowledge support from NASA grant NNX11AI52G. Authors thank all principal investigators and their staffs for providing ground- and
aircraft-based $NO_2$ measurements during the DISCOVER-AQ and KORUS-AQ campaigns.



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





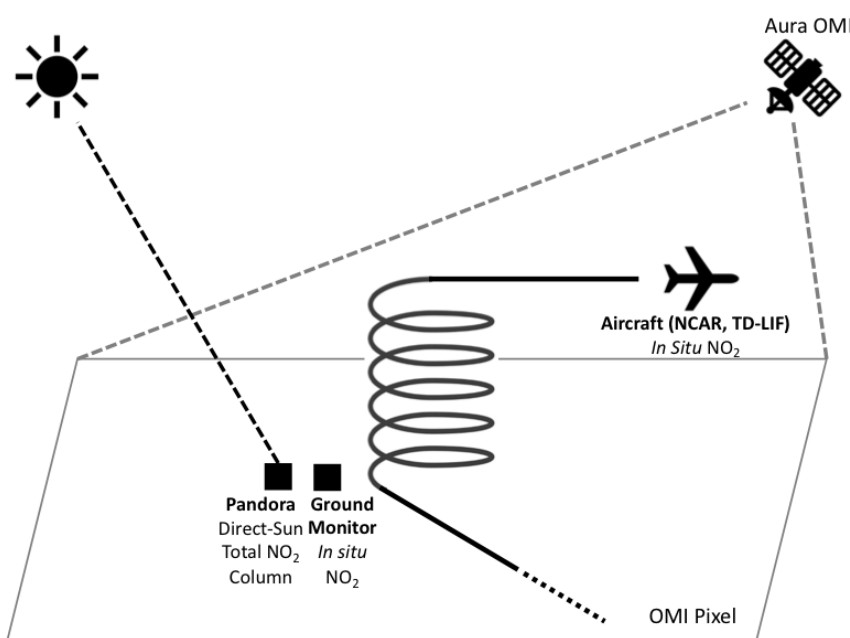

**Figure 1.** Conceptual illustration of NO$_2$ observations during the DISCOVER-AQ and KORUS-AQ field campaigns. The instruments used include ground-based monitors measuring in situ NO$_2$ volume mixing ratios, Pandora making direct-sun measurements to retrieve the total column NO$_2$, airborne instruments measuring in situ NO$_2$ profiles, and the Ozone Monitoring Instrument (OMI) aboard the Aura spacecraft reporting total and tropospheric columns NO$_2$.

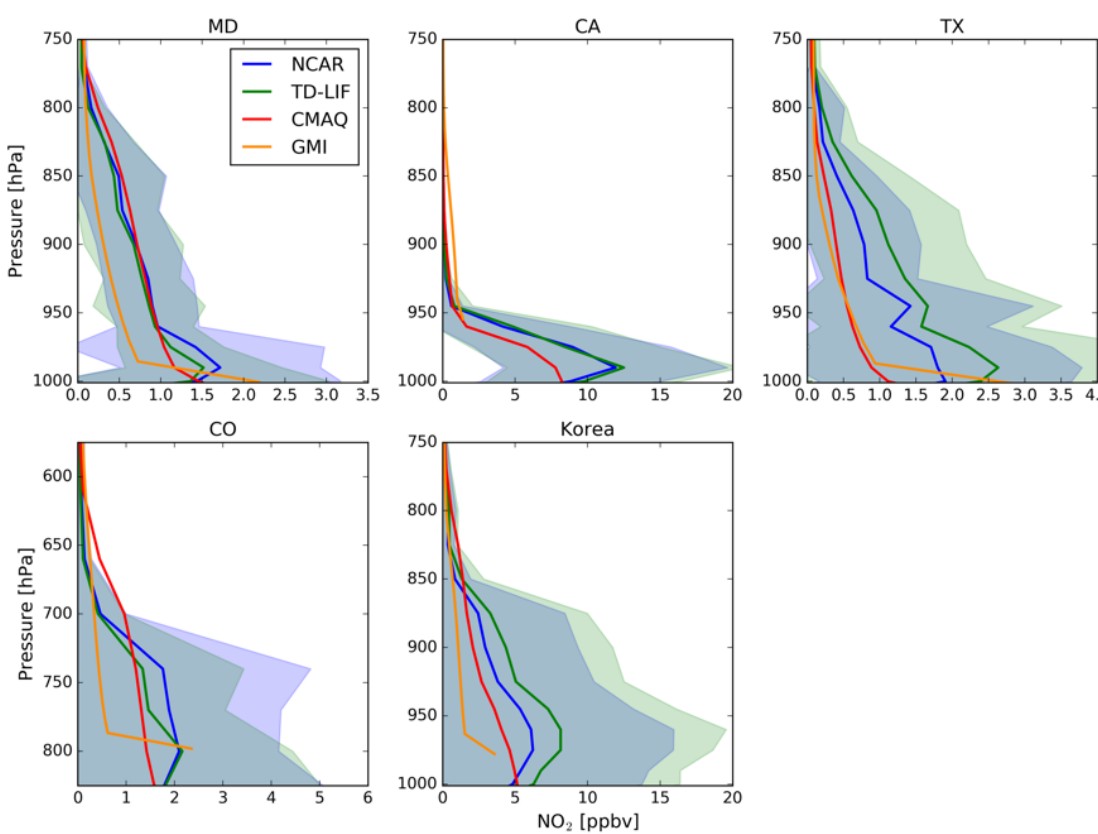

**Figure 2.** Mean early afternoon NO$_2$ profiles, both observed and modeled, for the DISCOVER-AQ and KORUS-AQ campaigns. Colored lines represent the average for airborne in situ profiles from NCAR (blue) and TD-LIF (green) instruments compared with simulated profiles from the GMI global model (orange) and the CMAQ (DISCOVER-AQ) or WRF-Chem (KORUS-AQ) regional models (red). The standard deviations of airborne profiles are indicated as shaded areas for NCAR (lavender) and TD-LIF (green) instruments. The blue-gray color represents the overlap of the two.





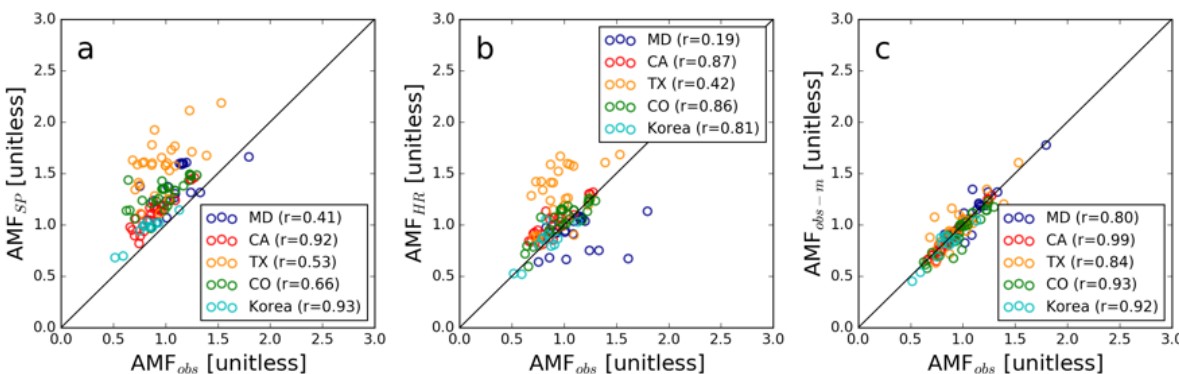

**Figure 3.** Comparison of AMFs calculated using observed $NO_2$ profiles ($AMF_{obs}$) with tropospheric AMFs in the OMI standard product ($AMF_{SP}$, 3a), and those calculated using $NO_2$ profiles from high-resolution model simulations ($AMF_{HR}$, 3b). The right panel (3c) compares tropospheric AMFs using daily versus campaign-average profiles ($AMF_{obs-m}$). The symbols are color-coded by campaign locations.





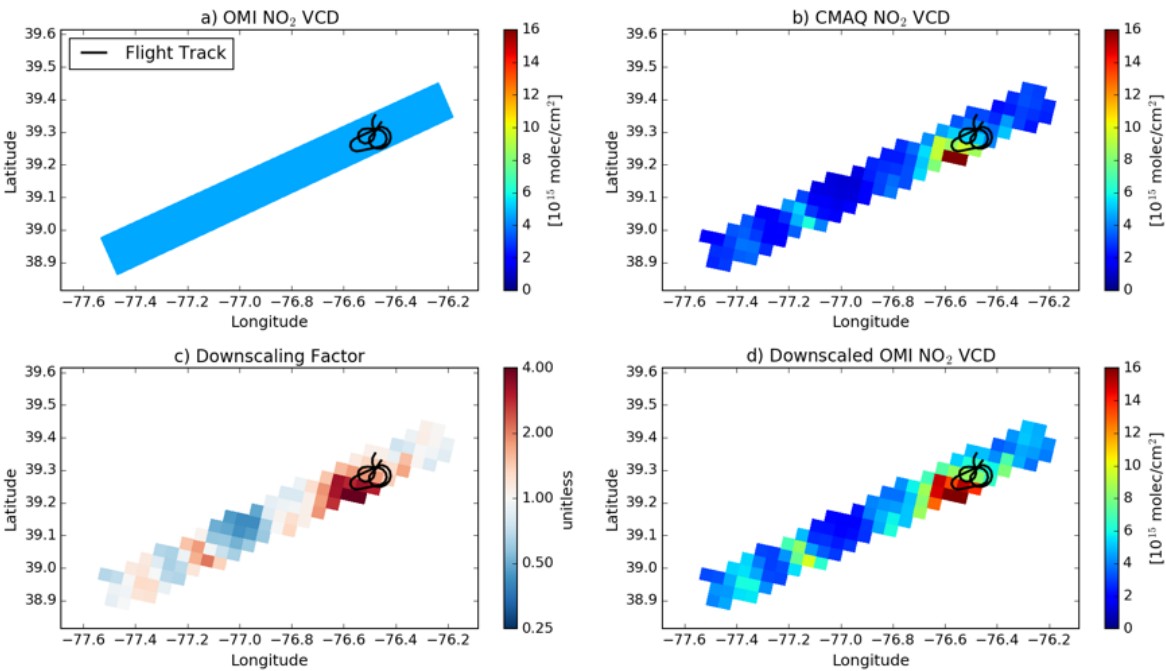

**Figure 4.** An illustration of downscaled OMI NO$_2$ for an OMI pixel over MD from orbit 37024 on July 1, 2011. Shown are the original OMI tropospheric NO$_2$ VCD (a), coincidently sampled CMAQ NO$_2$ VCD at a spatial resolution of 4×4 km$^2$ (b), the spatial weighting kernel (c), and downscaled OMI tropospheric NO$_2$ VCD (d). These pixels coincide with an airborne in situ NO$_2$ profile sampled during the DISCOVER-AQ Maryland campaign, and the flight route is marked with a black line.



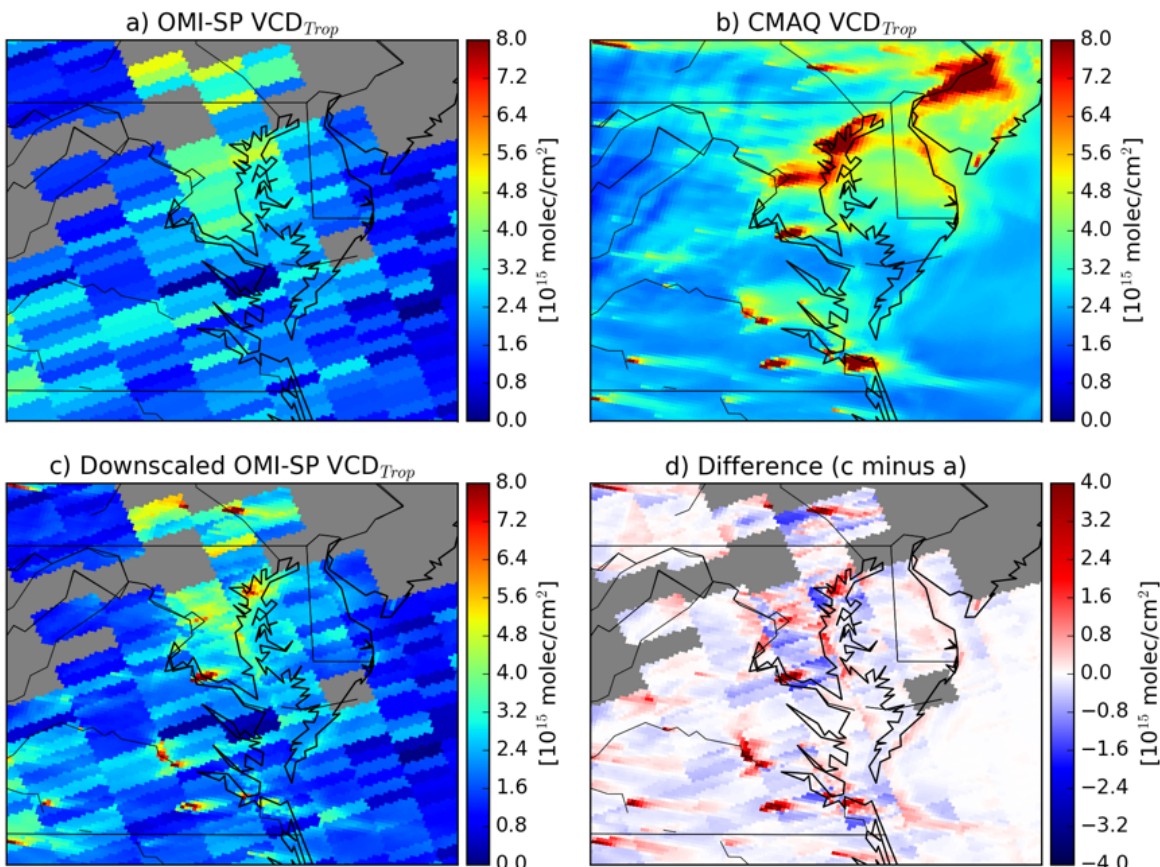

**Figure 5.** Tropospheric $NO_2$ VCD maps from (a) OMI SP, (b) CMAQ, and (c) downscaled OMI on July 29, 2011. The panel (d) shows the difference between downscaled and standard tropospheric $NO_2$ VCD data (c minus a). The gray areas represent pixels with effective cloud fraction $> 0.3$.





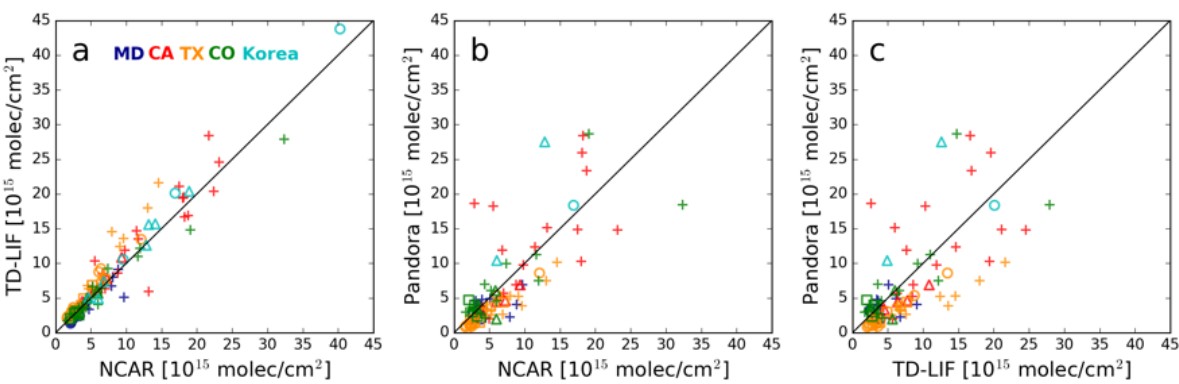

**Figure 6.** Comparison of NO$_2$ tropospheric columns derived from NCAR, TD-LIF, and Pandora instruments. Different colors represent the campaign location, and the symbols represent the type of surface monitors (open circle: photolytic converter, plus: molybdenum converter, triangle: CAPS, and square: CRDS).





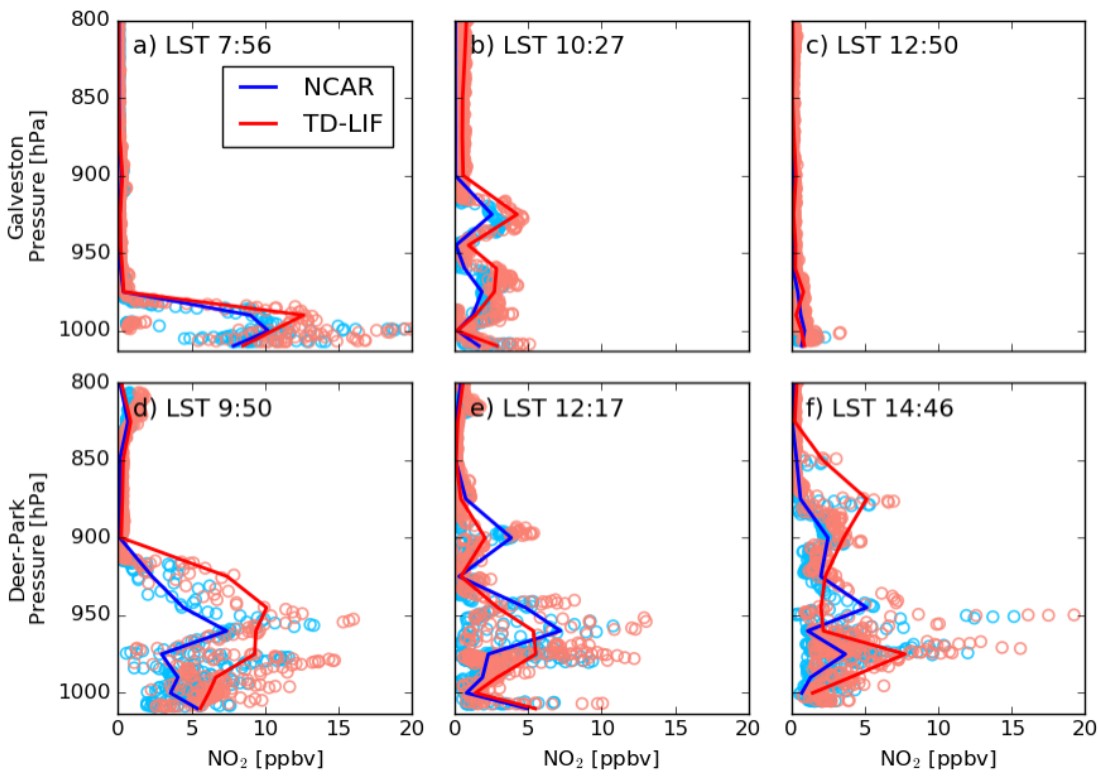

**Figure 7.** Vertical distribution of NO₂ mixing ratios at different local solar time (LST) over Galveston (top) and Deer-Park (bottom) in TX measured by the NCAR (light blue) and TD-LIF (orange) instruments. The circles in lighter colors represent 1-second measurements, and the solid lines show the mean values for NCAR (blue) and TD-LIF (red).

**Figure 8.** Comparison of tropospheric $NO_2$ columns from OMI with the data from NCAR (a, d, g), TD-LIF (b, e, h), and Pandora (c, f, i) instruments. OMI retrievals are performed using the default GMI (a-c), observed $NO_2$ profiles (d-i), or are downscaled (g, h, i) using a high-resolution (CMAQ/WRF-Chem) model simulations. Different colors represent the campaign locations.





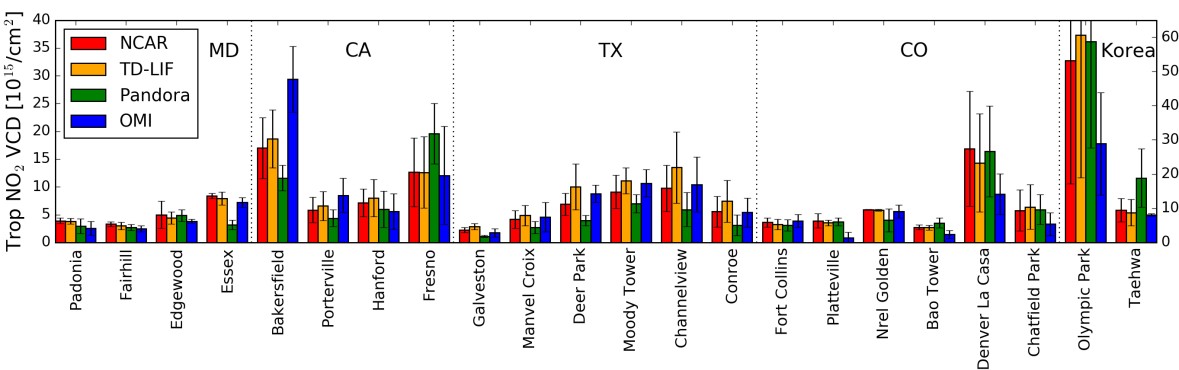

**Figure 9.** Site mean tropospheric NO₂ VCDs calculated from NCAR (blue), TD-LIF (orange), Pandora (green), and OMI (blue) instruments. The OMI data are derived using observed NO₂ profiles and downscaled using high-resolution model simulations.The vertical bars represent the standard deviations. A different y scale (on right, 0-65×10$^{15}$ molec./cm$^2$) is used for Korea.



**Table 1.** Campaign locations and time periods.

| Campaign | Location | Time period | Flight days |
| --- | --- | --- | --- |
| DISCOVER-AQ | Baltimore, Maryland | June–July 2011 | 14 |
| DISCOVER-AQ | San Joaquin Valley, California | January–February 2013 | 11 |
| DISCOVER-AQ | Houston, Texas | September 2013 | 10 |
| DISCOVER-AQ | Denver–Ft. Collins, Colorado | July–August 2014 | 15 |
| KORUS-AQ | Republic of Korea (South Korea) | May–June 2016 | 22 |



**Table 2.** Summary of ground supersites during DISCOVER-AQ and KORUS-AQ campaigns with ground based NO$_2$ measurements. The symbol N represents the sample size for aircraft and Pandora (in parentheses) measurements that are collocated with OMI observations. Surface NO$_2$ monitors include NO$_x$ analyzers with molybdenum converters (MC), NO$_x$ analyzers with photolytic converters (PC), Cavity Attenuated Phase Shift (CAPS), and Cavity Ring-Down Spectroscopy (CRDS).

| Campaign | Site | Latitude/Longitude | Elevation (m) | N | Ground monitor type |
|---|---|---|---|---|---|
| MD | Padonia | 39.46°N, 76.63°W | 120 | 6 (4) | PC |
| | Fairhill | 39.7°N, 76.86°W | 109 | 3 | PC |
| | Edgewood | 39.4°N, 76.47°W | 9 | 6 (5) | PC |
| | Essex | 39.31°N, 76.3°W | 13 | 3 (2) | MC |
| | Chesapeake* | 39.16°N, 76.34°W** | - | 3 (0) | - |
| CA | Bakersfield | 35.33°N, 119.0°W | 117 | 5 (3) | MC |
| | Porterville | 36.03°N, 119.06°W | 141 | 5 | CAPS |
| | Hanford | 36.32°N, 119.64°W | 80 | 7 (6) | MC |
| | Fresno | 36.79°N, 119.77°W | 97 | 8 | MC |
| TX | Galveston | 29.25°N, 94.86°W | 0 | 7 | PC |
| | Manvel Croix | 29.52°N, 95.39°W | 18 | 6 | CRDS |
| | Deer Park | 29.67°N, 95.13°W | 6 | 4 | MC |
| | Moody Tower | 29.72°N, 95.34°W | 64 | 4 (2) | PC |
| | Channelview | 29.80°N, 95.13°W | 6 | 4 | MC |
| | Conroe | 30.35°N. 95.43°W | 67 | 3 | MC |
| CO | Fort Collins | 40.59°N, 105.14°W | 1577 | 3 (2) | MC |
| | Platteville | 40.18°N, 104.73°W | 1522.5 | 5 (4) | MC |
| | NREL-Golden | 39.74°N, 105.18°W | 1846 | 4 (2) | CAPS |
| | Bao Tower | 40.04°N, 105.01°W | 1590 | 4 | CRDS |
| | Denver La Casa | 39.78°N, 105.01°W | 1602 | 5 (4) | MC |
| | Chatfield Park | 39.53°N, 105.07°W | 1675 | 5 | MC |
| Korea | Olympic Park | 37.52°N, 127.124°E | 26 | 4 (3) | PC |
| | Taehwa | 37.31°N, 127.311°E | 160 | 7 (2) | CAPS |

*Only aircraft spirals were performed over this site.

**The coordinate is approximate.

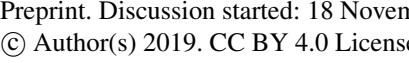

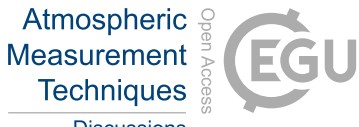

**Table 3.** Comparison between NCAR, TD-LIF, and Pandora $NO_2$ observations.

| Campaign | NCAR vs. TD-LIF Mean Diff. (%) (TD-LIF – NCAR) | r | NCAR vs. Pandora Mean Diff. (%) (Pandora – NCAR) | r | TD-LIF vs. Pandora Mean Diff. (%) (Pandora – TD-LIF) | r |
|---|---|---|---|---|---|---|
| MD | -9.6 | 0.87 | -24.5 | 0.42 | -18.3 | 0.18 |
| CA | 7.2 | 0.93 | 11.1 | 0.65 | 4.8 | 0.58 |
| TX | 31.9 | 0.97 | -39.1 | 0.94 | -53.9 | 0.93 |
| CO | -6.6 | 0.99 | -2.8 | 0.81 | 4.2 | 0.78 |
| Korea | 11.6 | 0.99 | 20.3 | 0.95 | 7.5 | 0.94 |

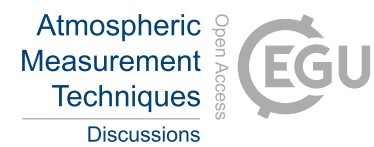

**Table A1.** Model options for each simulation. Note that all model options listed are for the domain used for the analysis.

| | MD | CA | TX | CO | Korea |
|---|---|---|---|---|---|
| Dates | 5/24/2011–8/1/2011 | 1/10/2013–2/28/2013 | 8/18/2013–10/1/2013 | 7/9/2014–08/20/2014 | 5/1/2016–5/31/2016 |
| **WRF Model Options** | | | | | |
| Version | 3.3 | 3.8 | 3.6.1 | 3.8.1 | 3.6.1 |
| Model Top | 100 hPa | 50 hPa | 50 hPa | 50 hPa | 50 hPa |
| Spatial Resolution | 4 km | 4 km | 4 km | 4 km | 4km |
| Vertical Levels | 34 | 35 | 45 | 37 | 52 |
| Radiation | LW: RRTM SW: Goddard | LW: RRTMG SW: RRTMG | LW: RRTM SW: Goddard | LW: RRTMG SW: RRTMG | LW: RRTM SW: Goddard |
| Land Surface Model | Noah Land Surface Model (Tewari et al., 2004) | Pleim-Xiu (Pleim and Xiu, 2003) | Pleim-Xiu (Pleim and Xiu, 2003) | Unified Noah Land Surface Model | Unified Noah Land Surface Model |
| Boundary Layer | YSU (Hong et al., 2006) | ACM2 (Pleim, 2007) | ACM2 (Pleim, 2007) | YSU (Hong et al., 2006) | MJY scheme |
| Meteor. Init. and Bound. Cond. | 12 km NAM | 12 km NAM | 12 km NAM | NCAR ECMWF | 0.25 degree GFS |
| **CMAQ Model Options** | | | | | **WRF-Chem** |
| Version | 5.0 | 5.2 | 5.0.2 | 5.2 beta | 3.6.1 (modified) |
| Coupled? | No | Yes | No | No | Yes |
| Chemical Mechanism | Carbon Bond (CB05) (Yarwood et al., 2005) | Carbon Bond (CB06, e51) | Carbon Bond (CB05) (Yarwood et al., 2005) | Carbon Bond (CB06, r3) | Reduced hydrocarbon (Pfister et al., 2014) |
| Aerosol | AE5 | AERO6 | AE5 | AERO6 | MOSAIC 4 bin |
| Chem. Init. and Bound. Cond. | 12 km CMAQ v5.0 simulation | 12-km CMAQ v5.2 simulation | MOZART (Outer domain) | RAQMS (Outer domain) | 24 km MACC for chemistry |
| Emissions | Described in Loughner et al. (2014) | 4km 2013 emissions (based off the 2011 NEI with year-specific updates to EGU point sources (CEMs data), fires and mobile emissions (MOBILE6) | 2012 TCEQ anthropogenic emissions Biogenic emissions Biogenic Emission Inventory System (BEIS) calculated within CMAQ | Described in report: https://www.colorado.gov/airquality/tech_doc_repository.aspx?action=open&file=FRAPPE-NCAR_Final_Report_July2017.pdf | Described in Goldberg et al. (2019) and Saide et al. (in preperation) |

LW: Long Wave, SW: Short Wave, RRTM: Rapid Radiative Transfer Model, RRTMG: Rapid Radiative Transfer Model for General Circulation Models, AE5: Aerosols with aqueous extensions version 5, MOZART: Model for OZone and Related chemical Tracers, RAQMS: Real Time Air Quality Monitoring System, MACC: Monitoring Atmospheric Composition and Climate



**Table A2.** Summary of NO$_2$ comparison between OMI Standard Product (OMI$_{SP}$) and NCAR, TD-LIF, and Pandora observations. The mean difference is calculated as OMI minus observations.

| Campaign | NCAR vs. OMI$_{SP}$ | | TD-LIF vs. OMI$_{SP}$ | | Pandora vs. OMI$_{SP}$ | |
|---|---|---|---|---|---|---|
| | Mean Diff. (%) | r | Mean Diff. (%) | r | Mean Diff. (%) | r |
| MD | -40.7 | 0.39 | -34.4 | 0.54 | -21.8 | 0.21 |
| CA | -53.8 | 0.77 | -56.9 | 0.81 | -58.5 | 0.24 |
| TX | -54.9 | 0.65 | -65.8 | 0.56 | -26.9 | 0.65 |
| CO | -67.5 | 0.73 | -65.2 | 0.75 | -68.2 | 0.72 |
| Korea | -41.9 | 0.87 | -47.9 | 0.87 | -60.1 | 0.8 |



**Table A3.** Same as A2, but for OMI using $AMF_{obs}$ ($OMI_{obs}$).

| Campaign | NCAR vs. $OMI_{obs}$ | | TD-LIF vs. $OMI_{obs}$ | | Pandora vs. $OMI_{obs}$ | |
|---|---|---|---|---|---|---|
| | Mean Diff. (%) | r | Mean Diff. (%) | r | Mean Diff. (%) | r |
| MD | -23.7 | 0.61 | -17.6 | 0.7 | 2.4 | 0.3 |
| CA | -42.4 | 0.73 | -45.8 | 0.75 | -47.9 | 0.2 |
| TX | -25.5 | 0.82 | -41.3 | 0.76 | 21.6 | 0.81 |
| CO | -54.2 | 0.7 | -50.5 | 0.71 | -55.2 | 0.69 |
| Korea | -33.9 | 0.87 | -39.2 | 0.86 | -53.3 | 0.79 |





**Table A4.** Same as A2, but for $OMI_{obs}$ with downscaling ($OMI_{DS}$).

| Campaign | NCAR vs. $OMI_{DS}$ | | TD-LIF vs. $OMI_{DS}$ | | Pandora vs. $OMI_{DS}$ | |
| :---: | :---: | :---: | :---: | :---: | :---: | :---: |
| | Mean Diff. (%) | r | Mean Diff. (%) | r | Mean Diff. (%) | r |
| MD | -24.1 | 0.75 | -18.0 | 0.85 | 0.8 | 0.31 |
| CA | 14.2 | 0.47 | 7.6 | 0.56 | 4.6 | 0.22 |
| TX | 9.5 | 0.94 | -13.8 | 0.91 | 78.3 | 0.93 |
| CO | -42.4 | 0.7 | -37.7 | 0.71 | -42.4 | 0.67 |
| Korea | -32.8 | 0.73 | -38.4 | 0.73 | -52.1 | 0.48 |