# Peer review of "Assessment of NO2 observations during DISCOVER-AQ and KORUS-AQ field campaigns"

_Atmospheric Measurement Techniques, 2019_

## Referee Comment (RC1) · Anonymous Referee #2 · 5 Dec 2019

General Comments

This paper assesses NO2 measurements from two field campaigns for their usefulness in interpreting satellite remote sensing observations. The paper focuses on what these field campaigns can tell us about how spatial resolution and a priori NO2 profiles affect interpretation of satellite observations. These are important and actively researched questions in the satellite retrieval community. The paper is well written, and the analysis is generally well supported. I have a couple questions listed below, but otherwise I recommend this paper for publication.

Specific comments

[Figure]

In section 2.1.2, the authors describe how aircraft profiles are linearly interpolated to the surface to meet the surface monitoring measurements. Figure 2 indicates that the merged aircraft profiles decrease towards the surface, while both models show increasing $NO_2$ towards the surface. It seems that the decrease towards the surface in the merged profiles is due to the interpolation – is there concern that the merged profile decreases towards the surface (which is generally unexpected over polluted areas), or that it has such a different shape than the model? AMF calculations for the satellite observations are sensitive to the profile shape in the lower troposphere, so what impact does this interpolation have on AMF calculations?

I was surprised to see the results discussed in the paragraph beginning line 304, which showed that monthly mean profiles capture the local variability as well as the daily profiles. Using monthly means rather than daily profiles would simplify future retrievals, so I would be interested to hear whether the authors think that this result is particular to the observed locations and seasons or if it can be applied more generally. For example, the only winter observations are in California (which isn't really that wintery), so could monthly mean profiles still be useful in these and other cases?

In section 3.2, the authors derive tropospheric columns from Pandora measurements by subtracting the OMI stratospheric column from the Pandora total column. I wonder whether this approach may be partially responsible for biases between Pandora and the aircraft observations. As a space-based instrument, OMI is more sensitive to stratospheric $NO_2$, while Pandora has a greater tropospheric sensitivity. How do the authors account for differences in vertically-resolved sensitivity between the instruments? Also, what is the possible effect of subpixel variability in stratospheric $NO_2$ within the OMI pixel? There's nothing in the discussion that describes potential errors in the stratosphere-troposphere separation.

Lastly, there's a recent paper by Judd et al (https://doi.org/10.5194/amt-12-6091-2019) that may be of interest to the authors (I am not affiliated with this paper, but thought it was relevant).

---

## Referee Comment (RC2) · Anonymous Referee #3 · 25 Feb 2020

Overview

The paper by Choi et al. compares aircraft (from several campaigns) measurements from two different instruments and ground-based PANDORA measurements to satellite observed NO2 columns from OMI. The paper aims to interpret the differences between the datasets and explores different techniques how the satellite observations can be compared to aircraft measurements. The authors look into the effect of using a more accurate NO2 a priori profile and the effect of correcting for the large satellite pixel size. Overall, I found the paper was well-written and scientifically sound, it fits well in the scope of Atmospheric Measurement Techniques. I have some suggestions (listed

below), after addressing these, I would recommend the paper for publication in AMT.

Specific Comments

Abstract and Introduction: It would be helpful to include the time period of the campaigns in the abstract and introduction, just roughly, e.g. "...these campaigns took place between 2011 and 2016."

p. 9, l. 254: Include what filters have been applied to the OMI data. Here, you mention the row anomaly, but what other filters? I think it would be good to mention it here, instead of l. 369 "quality-controlled, cloud-free". What do you mean specifically with "quality-controlled"?

The correlation is discussed for each location (in Sects. 2.3.2 and 3.2); I think that there might not be enough measurements for many of these for the correlation to be meaningful. Some of the high correlation appears for locations where there are only a handful of measurements (e.g. Korea), with 1 or 2 high column amounts. I think it would be better to just discuss the correlation for all measurements from all locations rather than separating them. I think it's ok to talk about the differences for each location.

Table 3 (and A2-A4): Include the correlation and difference for all locations (total), also include columns for the sample size (N)- I know it is listed in Table 2, but it would be helpful to have this information all together in these tables. I believe from this it will be clear that maybe not too much weight should be given to the correlation for individual sites.

Where are the spirals relative to Pandora? Are the spirals flown over the Pandora locations? Or how far is the distance? It might be useful to show maps (could be in the appendix), one for each of the five sites with the location of the spirals, PANDORA site and maybe (if it's not too busy) the pixel outline from OMI observations used for the comparison. This will be helpful for the discussion and interpretation of the differences between the measurements taken on the aircraft, from the satellite and from the

ground.

Fig.5: Where is this? Please include lon/lat ticks, and the location name in the caption.

Fig. 9: The error bar is cut off from Olympic park. I would suggest changing the figure slightly (it is necessary to read the caption to know which colorbar applies, it is not immediately obvious to the reader): Maybe instead of using two different y-axes, it might be better to include a scale break for the Olympic park columns, or a logarithmic scale.

Table 2: It says in the caption "Pandora in parenthesis", many do not have a number in parenthesis, is that if the spirals are the same number as Pandora, or are no Pandora measurements available? This is not clear. To make it easier for the reader include parenthesis with the number of PANDORA measurements everywhere.

Did you evaluate the effect of changing the time difference slightly to a stricter or more relaxed criteria to 1.5h (1h or 30min, or 2h)? Are some of the outliers related to a large time difference?

I guess you only considered the OMI pixel that overlapped with the aircraft data, or did you average the OMI observations? Was this specifically mentioned somewhere? It should be explained.

Technical Corrections

p.1 l.15: "very well, but. . ."

p.2 l.18 "but mostly to OMI's areal (>312km2) averaging" change to "but mostly to OMI's large footprint (>312 km2)."

p. 5 l.138: "figures below" change to "Figs. X-Y." (Which figures? Please specify.)

p. 9, l. 252: Space is missing ". Since"

---

## Author Comment (AC1) · 24 Mar 2020

**We thank the reviewer #2 for his/her helpful comments that have helped improve the manuscript. Below, please find our responses in boldface for each comment. We have updated the manuscript addressing these comments.**

General Comments

This paper assesses NO2 measurements from two field campaigns for their usefulness in interpreting satellite remote sensing observations. The paper focuses on what these field campaigns can tell us about how spatial resolution and a priori NO2 profiles affect interpretation of satellite observations. These are important and actively researched questions in the satellite retrieval community. The paper is well written, and the analysis is generally well supported. I have a couple questions listed below, but otherwise I recommend this paper for publication.

Specific comments

1) In section 2.1.2, the authors describe how aircraft profiles are linearly interpolated to the surface to meet the surface monitoring measurements. Figure 2 indicates that the merged aircraft profiles decrease towards the surface, while both models show increasing NO2 towards the surface. It seems that the decrease towards the surface in the merged profiles is due to the interpolation – is there concern that the merged profile decreases towards the surface (which is generally unexpected over polluted areas), or that it has such a different shape than the model? AMF calculations for the satellite observations are sensitive to the profile shape in the lower troposphere, so what impact does this interpolation have on AMF calculations?

**Main idea of Figure 2 is to show the nature of variability in observed and simulated NO2 vertical profiles over the entire domain for each individual campaign. Due to mismatch in both spatial and temporal sampling between models and observations, this figure is merely for a qualitative understanding, and cannot be directly linked to results presented in the later sections. Our analysis focusing on comparison of different data sets and examining the AMF effect uses more restrictive collocation (spatial and temporal). To address this concern, we have added the following statements (Page 3, Line 124):**

> **Figure 2 also shows the nature of variability in observed and simulated NO2 vertical profiles over the campaign domains. The observed differences between the model and observations arise primarily from mismatch in both spatial and temporal sampling. Use of more restrictive collocation (spatial and temporal) applied for comparing different data sets in Section 3.1 and examining the AMF effect in Section 2.3.2 would have resulted in different vertical distributions.**

**Moreover, the linear interpolation between the surface and lowest aircraft altitude (~250 m) is done to partially account for missing information. The missing information certainly represents a significant source of errors, and the interpolation adopted here is our attempts to mitigate some of these errors. We have discussed these limitations in Section 3.4 (Page 14, Line 424-432)**

2) I was surprised to see the results discussed in the paragraph beginning line 304, which showed that monthly mean profiles capture the local variability as well as the daily profiles. Using monthly means rather than daily profiles would simplify future retrievals, so I would be interested to hear whether the authors think that this result is particular to the observed locations and seasons or if it can be applied more generally. For example, the only winter observations are in California (which isn't really that wintery), so could monthly mean profiles still be useful in these and other cases?

**We agree with the reviewer. AMFs calculated using local, daily vs campaign average monthly NO$_2$ profiles show a good agreement with correlation coefficient of 0.9 (Figure 3), suggesting that local monthly mean profile can be a reasonable choice as discussed in the manuscript. Use of monthly mean profiles also simplifies retrieval process as the reviewer pointed out. We have discussed this aspect in Section 2.3.2 (Page 11, Lines 315-323). However, previous works (e.g., Laughner et al, 2018) have shown that errors form using monthly average profiles are as large as 30%. This may suggest that daily profiles that do not capture plume can potentially lead to considerable errors. This is the topic of active research, and should be pursued as more campaign data become available.**

3) In section 3.2, the authors derive tropospheric columns from Pandora measurements by subtracting the OMI stratospheric column from the Pandora total column. I wonder whether this approach may be partially responsible for biases between Pandora and the aircraft observations. As a space-based instrument, OMI is more sensitive to stratospheric NO2, while Pandora has a greater tropospheric sensitivity. How do the authors account for differences in vertically-resolved sensitivity between the instruments? Also, what is the possible effect of subpixel variability in stratospheric NO2 within the OMI pixel? There's nothing in the discussion that describes potential errors in the stratosphere-troposphere separation.

**Thank you for pointing this out. Use of OMI stratospheric NO2 columns to derive tropospheric columns from Pandora could impact the comparison between Pandora and aircraft observations. But, this is likely a minor effect because (1) OMI stratospheric NO2 estimates are fairly accurate following improvements and assessment of slant column retrievals by the European and NASA teams (e.g., Marchenko et al., 2015; Zara et al., 2018; Boersma et al., 2018); (2) contribution of the stratosphere on total column NO2 is small in polluted areas, where these campaigns took place; (3) the stratospheric NO2 field is relatively smooth and is unlikely to vary significantly within few kilometers; and (4) variation in vertical sensitivity for both OMI and Pandora measurements are accounted for in the retrievals. We have included the following statement in the revised manuscript (Page 12, Line 364).**

> **The use of OMI stratospheric NO$_2$ columns to derive tropospheric columns from Pandora could impact the comparison between Pandora and aircraft observations; this approach is unlikely to be a significant factor over the polluted DISCOVER-AQ and KORUS-AQ campaign domains.**

4) Lastly, there's a recent paper by Judd et al (https://doi.org/10.5194/amt-12-6091-2019) that may be of interest to the authors (I am not affiliated with this paper, but thought it was relevant).

**Thanks for the suggestion. We have cited Judd et al in the revised version.**

---

## Author Comment (AC2) · 24 Mar 2020

**We thank the reviewer #3 for his/her helpful comments that have helped improve the manuscript. Below, please find our responses in boldface for each comment. We have updated the manuscript addressing these comments.**

Overview
The paper by Choi et al. compares aircraft (from several campaigns) measurements from two different instruments and ground-based PANDORA measurements to satellite observed NO2 columns from OMI. The paper aims to interpret the differences between the datasets and explores different techniques how the satellite observations can be compared to aircraft measurements. The authors look into the effect of using a more accurate NO2 a priori profile and the effect of correcting for the large satellite pixel size. Overall, I found the paper was well-written and scientifically sound, it fits well in the scope of Atmospheric Measurement Techniques. I have some suggestions (listed below), after addressing these, I would recommend the paper for publication in AMT.

Specific Comments
1) Abstract and Introduction: It would be helpful to include the time period of the campaigns in the abstract and introduction, just roughly, e.g. "...these campaigns took place between 2011 and 2016."

**We have specified the years of campaigns in the abstract as suggested.**

2) p. 9, l. 254: Include what filters have been applied to the OMI data. Here, you mention the row anomaly, but what other filters? I think it would be good to mention it here, instead of l. 369 "quality-controlled, cloud-free". What do you mean specifically with "quality-controlled"?

**We agree. We have added the following sentence in the revised manuscript, and removed similar statement in line 369 of the previous version:**

> **We use OMI pixels with cloud radiance fraction less than 50 % and quality flags suggesting good data.**

3) The correlation is discussed for each location (in Sects. 2.3.2 and 3.2); I think that there might not be enough measurements for many of these for the correlation to be meaningful. Some of the high correlation appears for locations where there are only a handful of measurements (e.g. Korea), with 1 or 2 high column amounts. I think it would be better to just discuss the correlation for all measurements from all locations rather than separating them. I think it's ok to talk about the differences for each location.

**We have modified the statements discussing correlations of individual campaigns with those with overall correlation, as suggested. The sentence in Section 2.3.2 (Page 10, Line 304) now reads:**

The correlation ranges from fair (r=0.41, N=21) for MD and TX to excellent (r>0.9, N=36) for CA and Korea with the overall correlation coefficient of 0.53.

And the sentence in Section 3.2 (Page 12, Line 367) now reads:

The overall correlation coefficient between Pandora and the airborne NCAR and TD-LIF measurements are 0.94 and 0.91, respectively, with higher correlation in CO, TX, and Korea and lower correlation in MD and CA.

4) Table 3 (and A2-A4): Include the correlation and difference for all locations (total), also include columns for the sample size (N)- I know it is listed in Table 2, but it would be helpful to have this information all together in these tables. I believe from this it will be clear that maybe not too much weight should be given to the correlation for individual sites.

We have revised the tables as suggested.

5) Where are the spirals relative to Pandora? Are the spirals flown over the Pandora locations? Or how far is the distance? It might be useful to show maps (could be in the appendix), one for each of the five sites with the location of the spirals, PANDORA site and maybe (if it's not too busy) the pixel outline from OMI observations used for the comparison. This will be helpful for the discussion and interpretation of the differ- ences between the measurements taken on the aircraft, from the satellite and from the ground

Thank you for pointing that out. The surface sites hosting in-situ $NO_2$ monitors and Pandora instruments are within 2-5 km from aircraft spirals during DISCOVER-AQ and 10-20 km from aircraft ascents/descents during the KORUS-AQ field campaigns. During DISCOVER-AQ, the spiral diameter was about 4km on average. We have included this information in Section 2.1 (Page 4, Line 87) in the revised manuscript as below:

The P-3B aircraft made spirals of ~4km diameter whereas the DC-8 ascents/descents covered 10-20 km. Consequently, the distance between the ground and aircraft locations was 0-5 km during the DISCOVER-AQ and 10-20 km during the KORUS-AQ campaign.

We have also revised Fig. 4 showing the location of a surface site as an example to provide the idea of relative locations of ground, aircraft, and OMI measurements.

6) Fig.5: Where is this? Please include lon/lat ticks, and the location name in the caption.

This is for the Maryland campaign. We have revised the figure and the figure caption as suggested.

7) Fig. 9: The error bar is cut off from Olympic park. I would suggest changing the figure slightly (it is necessary to read the caption to know which colorbar applies, it is not immediately obvious to the reader): Maybe instead of using two different y-axes, it might be better to include a scale break for the Olympic park columns, or a logarithmic scale.

**We have revised Fig. 9 using a Y scale break as suggested.**

8) Table 2: It says in the caption "Pandora in parenthesis", many do not have a number in parenthesis, is that if the spirals are the same number as Pandora, or are no Pandora measurements available? This is not clear. To make it easier for the reader include parenthesis with the number of PANDORA measurements everywhere.

**We agree with the reviewer. The table is now modified for clarity.**

9) Did you evaluate the effect of changing the time difference slightly to a stricter or more relaxed criteria to 1.5h (1h or 30min, or 2h)? Are some of the outliers related to a large time difference?

**We tested the effect of different time windows on the results. Narrower time windows improve temporal matching, but decrease the number of samples. Using wider time windows could help reduce the impact of outliers and found to slightly decrease the correlations, but the change is marginal. Our selected time window was intended to maximize the number of samples while reducing effects from diurnal variation of NO$_2$ profiles. We have added the following statement in the revised manuscript (Page 4, Line 114):**

> **This time window of ±1.5 hour is selected to maximize the number of samples while reducing effects from diurnal variation of NO$_2$.**

10) I guess you only considered the OMI pixel that overlapped with the aircraft data, or did you average the OMI observations? Was this specifically mentioned somewhere? It should be explained.

**We select only the OMI pixel that overlap with individual aircraft profiles. Changes made in Section 3.3 (Page 13, Line 377)**

Technical Corrections
   11) p.1 l.15: "very well, but. . ."

**Done.**

12) p.2 l.18 "but mostly to OMI's areal (>312km2) averaging" change to "but mostly to OMI's large footprint (>312 km2)."

**Done. Changed as suggested.**

13) p. 5 l.138: "figures below" change to "Figs. X-Y." (Which figures? Please specify.)

**Changed as suggested.**

14) p. 9, l. 252: Space is missing ". Since"

**Done, thanks.**